# Skewed logit model for analyzing correlated infant morbidity data

**Ngugi Mwenda** [1]*, **Ruth Nduati**[2], **Mathew Kosgei**[1], **Gregory Kerich**[1]

**1** School of Science and Aerospace Studies, Department of Mathematics, Physics and Computing, Moi University, Eldoret, Kenya, **2** Department of Paediatrics, University of Nairobi, Nairobi, Kenya

* samwenda87@gmail.com

## Abstract

### Background

Infant morbidity is a topic of interest because it is used globally as an indicator of the status of health care in a country. A large body of evidence supports an association between bacterial vaginosis (BV) and infant morbidity. When estimating the relationship between the predictors and the estimated variable of morbidity severity, the latter exhibits imbalanced data, which means that violation of symmetry is expected. Two competing methods of analysis, that is, (1) probit and (2) logit techniques, can be considered in this context and have been applied to model such outcomes. However, these models may yield inconsistent results. While non-normal modeling approaches have been embraced in the recent past, the skewed logit model has been given little attention. In this study, we exemplify its usefulness in analyzing imbalanced longitudinal responses data.

### Methodology

While numerous non-normal methods for modeling binomial responses are well established, there is a need for comparison studies to assess their usefulness in different scenarios, especially under a longitudinal setting. This is addressed in this study. We use a dataset from Kenya about infants born to human immunodeficiency virus (HIV) positive mothers, who are also screened for BV. We aimed to investigate the effect of BV on infant morbidity across time. We derived a score for morbidity incidences depending on illnesses reported during the month of reference. By adjusting for the mother's BV status, the child's HIV status, sex, feeding status, and weight for age, we estimated the standard binary logit and skewed logit models, both using Generalized Estimating Equations.

### Results

Results show that accounting for skewness in imbalanced binary data can show associations between variables in line with expectations documented by the literature. In addition, an in-depth analysis accounting for skewness has shown that, over time, maternal BV is associated with multiple health conditions in infants.

**Data Availability Statement:** Data are available at https://osf.io/yfzw5.

**Funding:** The author(s) received no specific funding for this work.

**Competing interests:** The authors have declared that no competing interests exist.

## Interpretation

Maternal BV status was positively associated with infant morbidity incidences, which highlights the need for early intervention in cases of HIV-infected pregnant women.

## Introduction

Skewed and non-normal data are commonly observed in health research. Usually, the dataset is transformed, censored, or truncated to impose normality, rather than modeling the data in its natural state [1]. Many conventional approaches to modeling lead to incorrect estimates of parameters and standard errors due to the assumptions imposed.

For example, imbalances can occur in binary response data, when symmetry is violated. There are two types of models which are typically employed to analyze data in these scenarios —1) logit and 2) probit models. Logit models have error variables that follow a logistic distribution and this type of model is considered to be characteristic of discrete choice models. The probit model uses the cumulative standard normal distribution function and assumes the error term is normally distributed. Although this assumption is viewed as a reasonable compromise to achieve mathematical simplicity and parsimonious results, its suitability has been doubted of late.

Several recent studies have investigated various ways of handling non-normal data. However, few have focused on the methodology. For example, a paper published by Bono et al. [2] details several non-normal distributions typical in health, education, and social science, but their substantiation in the literature remains scarce. Further, several other distributions are not considered in this paper, suggesting that they were not common in the study's period of reference. However, these distributions could be vital in answering some important scientific questions on binary responses that suffer from substantial departure from the commonly assumed symmetric logistic distribution [3–5].

For example, for the Bernoulli distribution, binary asymmetry is defined as the sensitivity to changes in the independent variable that is not maximized at 0.5. This means that a stimulus in any of the independent variables for any individual with probability $P = 0.5$ is not exaggerated. Assuming symmetry in some settings could be inefficient and can lead to biased estimators [4].

The importance of normality and symmetry in traditional methods of data analysis cannot be under-estimated. There is a need for compromise between statistical simplicity and plausible estimates of parameters when these assumptions do not hold. Questions regarding the suitability of the assumption-based methods have been raised in the literature [4]. Put differently, although the numerous probability distribution function options can fit the data quite well, the data need to speak for themselves, rather than being forced into a model with assumptions [1].

There is mounting scientific evidence regarding the inconsistency and weakness of the logit and probit models for skewed binary response data. Recent studies have proposed alternative methods for handling binomial responses, such as: a gamma generated logistic distribution [6], gamma and log-normal distributions [7], improved analysis for skewed continuous responses [8], a skewed Weibull regression model [9], a generalized logistic distribution [10], and a skewed logit model [4]. This shows that modeling non-normality continues to be a topic of importance in recent general research. However, few methods have been considered and applied in health research. Most of the literature and applications have focused on cross-sectional data in social, political, and economics research [3, 11–15].

Our study is focused on (1) estimating skewness parameter from GLM (2) applying the value in a longitudinal study on infant morbidities under GEE [34].

Morbidity is the state of being symptomatic or unhealthy due to a disease or condition [16] and this can be experienced at any stage in life. This study is focused on BV related morbidities, since this remains a major point of concern globally and particularly in Africa, where the majority of BV cases are recorded [17–19]. Child morbidity and mortality as a consequence of BV in conjunction with human immunodeficiency virus (HIV) has been a significant hindrance to meeting goal three of the United Nations Sustainable Development Goals (UN-SDGs) on Good Health and Well-being [20], which aims to end preventable deaths of newborns and children under 5 years of age.

The scientific literature has established a link between BV and adverse outcomes in mothers and their children [21]. Past studies have investigated the occurrence of health deficiencies [22, 23], pregnancy loss, labor complications and preterm delivery [24–26], as well as spontaneous and recurrent abortions [27] among mothers, while others have reported adverse outcomes such as neonatal malformations [28] and low birth weight [29] among the babies. While some studies have tried to investigate the effects of BV in the context of HIV infection [23, 30, 31], there is still a lack of knowledge regarding the long-term effects in these cases.

To shed light on this topic, we apply the skewed logit model using Generalized Estimating Equations (GEE) to evaluate the variations in the data across time in months and thereby, better understand the infant morbidities. This approach relaxes the strong conditional probability on a binary response, thereby accommodating for the heterogeneity of repeated measures on the same subjects, and accounting for interaction effects in the selected covariates across time.

## Materials and methods

### Data

The Nairobi Infant Morbidity Study (NIMS) was a randomized clinical trial carried out by scholars in the International AIDS Research and Training Program supported by grant NICHD-23412 from the National Institutes of Health. The objective was to collect high quality longitudinal data on morbidity and mortality of babies from HIV-positive pregnant women in a random sample considering mothers who either breastfed or gave their baby formula. The description, analysis and findings of the original study can be found elsewhere [32].

The study participants were drawn from a population of 16529 pregnant mothers attending four antenatal clinics in Nairobi, Kenya. After screening for HIV, 2315 were found to be positive. Of these women, 425 were selected and verbally agreed to be enrolled in the study. At each prenatal visit, each woman was subjected to a standard physical and clinical examination, and an interview.

Before birth, at 32 weeks of pregnancy, pelvic examination, including analysis of vaginal and cervical secretions were conducted for each woman to determine their BV status. This was done using sterile Dacron swabs by a trained clinical officer and the Nugent criteria was used to qualify a woman for a BV diagnosis. A pH value from the swab, of $\geq 7$ was considered a case, indicating alkalinity of the vaginal fluids and inhibition of bad bacteria such as as *Trichomonas, Candida albicans, Enterobacteriaceae, Staphylococcus* and *Streptococcus*.

Immediately after birth, infants were assessed for HIV using enzyme-linked immunosorbent assay (ELISA). Those who tested positive were subjected to a more accurate Polymerase chain reaction (PCR) test. Infants who had three consecutive negative tests were deemed negative. The pairs of infants who survived were regularly re-examined over the next two years and their history of ailments were documented at every visit.

The study data was collected in two ways, scheduled and unscheduled visits. Scheduled visits meant that the dyad pairs were supposed to come to the clinic for examination at a specific time, while unscheduled visits meant they could pop in any time in case of an illness. Other physical examinations of the baby, including details like sex, weight, and height, were observed and recorded.

The planned visits were bi-weekly during the first 3 months and monthly thereafter for up to two years. In all scenarios, data were collected either through parental report or diagnosis at the hospital or clinic. Of the total number of women enrolled, complete records from birth to six months were only available for 401 women. The other 24 women either had miscarriages or still births or did not complete the follow up appointments. Of the 401 women, 74 pairs had missing values, either for the mother's BV measure or for the morbidity incidences of the infants. To address this, we applied a missing completely at random (MCAR) mechanism. There is sufficient evidence that, using the GEE approach, this approach still enables a consistent estimate of the regression parameters so long as the mean model is correctly specified [33].

A standard questionnaire developed by the principal investigator of the study to identify illnesses was completed for both the mother and the child. This was achieved using a 19-item yes/no morbidity questionnaire which purports to measure health status of an infant. The total score of the questionnaire is computed as the count of all the "yes" responses. There were a total of 1962 observations from 327 pairs of mothers and babies. From the total score, we created a binary response of; (1) those who did not have any illness and (2) those who had either minimal or severe illnesses. Table 1 presents an initial exploratory analysis used to identify the asymmetry in the total responses for each month. This evidence of asymmetry justifies the use of the skewed logit model.

## Ethical approval

The study protocol was approved by the institutional review boards of the University of Washington and University of Nairobi. Verbal consent was obtained from all mothers prior to their inclusion in the study. The investigators in the study did not require documentation of any consent for the participants because at the time of the study, written consent was not mandated by the ethics bodies involved. Therefore, at that time, no procedures regarding written consent were violated given the research context of doing the study in Kenya.

## Statistical model

Generalized Linear Models (GLMs) and the GEEs were used to model infant morbidity. The first modeling approach to determine the need for the skewed logit model and the value for skewness was carried out with GLMs [34]. The response variable was the health status of the

**Table 1. BV with morbidity incidences reported from month one to six for both BV-exposed and unexposed babies in the Nairobi data survey.**

| Time in Months | BV present(n = 148) | BV absent(n = 179) | Total(n = 327) |
|---|---|---|---|
| 1 | 115(78%) | 85(47%) | 200(61%) |
| 2 | 97(66%) | 84(47%) | 181(55%) |
| 3 | 97(66%) | 85(48%) | 182(56%) |
| 4 | 92(62%) | 101(56%) | 193(59%) |
| 5 | 86(58%) | 96(53%) | 182(56%) |
| 6 | 79(53%) | 103(58%) | 182(56%) |

infant within a particular month at the time of the hospital visit or the reported health status about the infant from the mother. For our study, we considered all health events, whereby a health event occurred if an infant was reported to have experienced any illness within the month.

Let $Y_{it}$ be the response for subject $i$ measured at different points in time $t = 1, \ldots, n_i$ denote the outcome vector for subjects $i = 1, 2, \ldots, N$ and $\mathbf{x_{it}}$ is a $n_i \times q$ matrix of covariate variables for subject $i$. The expected value is given by $E(Y_{it}) = \pi_{it}$ and the linear predictor that relates the mean to the covariates is given by

$$k(\pi_{it}) = \eta_{it} = \mathbf{x_{it}}^\top \boldsymbol{\beta} \tag{1}$$

where $\mathbf{x_{it}}$ is the covariate vector for subject $i$ at time $t$ with length $q$. This includes the infant weight, mother's BV status, HIV status of the infant, and feeding status of the infant. $k^{-1}(.)$ is a known link function such as the skewed logit model and $\boldsymbol{\beta}$ are regression parameters.

For each infant status of illness at any chosen time point, the response follows a Bernoulli distribution with $p_i$(probability of being ill=$\pi_{it}$) and is specified as:

$$Y_{it} \sim Bern(1, \pi_{it}) \tag{2}$$

To model the outcome, the logit and probit models are preferred options, but they both have conditional probability distributions, which have a maximum at 0, such that $P_i$ for $i \in (0, 1)$ is 0.5 and thus, they have a fixed symmetry of 0.5. However, this assumption of symmetry may not be realistic to all Bernoulli responses and therefore, not desired [4, 11, 15]. For this reason, the skewed logit approach is employed here, taking advantage of the fact that the logit model is nested within the skewed logit model (Fig 1). There are reported similarities in terms of model specification, estimation, and iterations. Using the skewed logit model made it possible to see if the data were skewed and therefore, to estimate the skewness value.

The probability of a child experiencing illness is given by

$$\Pr(\text{illness} = 1) = k^{-1}(\mathbf{x}_{it}^\top \boldsymbol{\beta}) \tag{3}$$

In this work, we aim to consider a response that violates the symmetry assumption, using the framework described above. Following Burr [35], $k^{-1}(.)$ accommodates asymmetry through;

$$k^{-1}(.) = \Pr(y_{it} = 1 \mid x_{it}) = 1 - \frac{1}{(1 + \exp(x_{it}\boldsymbol{\beta}))^\alpha} \tag{4}$$

for $\alpha > 0$ and this is the skew value to be estimated.

This variation implies that the maximum is no longer restricted to $P = 0.5$. Since the skew value cannot be observed, we fit a regression of all covariates under the skewed logit model using the GLM approach [34]. Further, we use the $\alpha$ obtained as a proxy for the disturbance to be used in the GEE [36].

To obtain robust standard errors that are meaningful for the parameter estimates, we adopted the Huber sandwich estimator [37, 38], which has the ability to relax the intra-group correlation. To increase the efficiency of model convergence, we specify a tolerance value of 0.0001 and set the maximum number of iterations to 100.

The applicability of the two models using the set of covariates was determined by the likelihood ratio test that compares the logit and the skewed logit model to identify any significant differences [39].

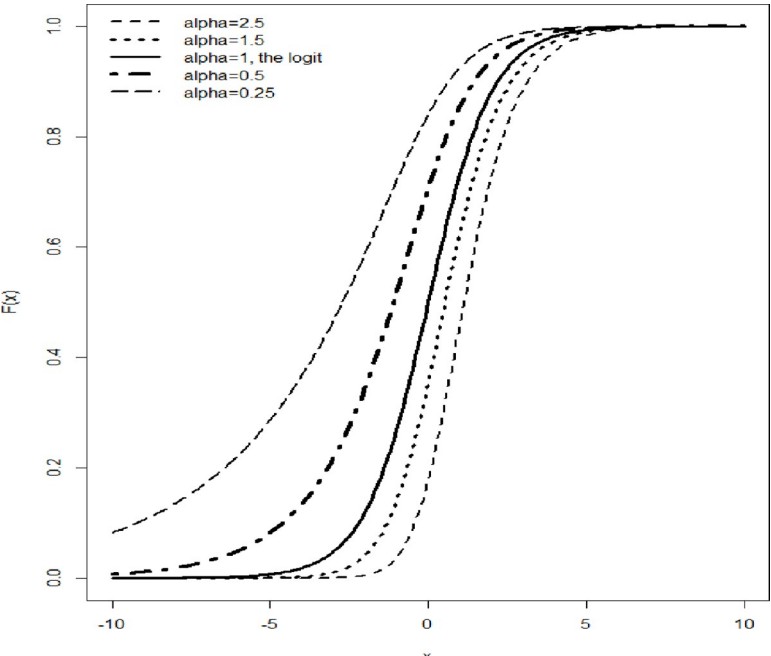

**Fig 1. Cumulative density function of the skewed logit model with different values of skewness.** The bold continuous line represents the logit model which assumes symmetry.

## Estimation of parameters using the GEE

Developed by Liang and Zeger [40] in their land mark paper, GEE can be used to model correlated data and give a marginal inference interpretation. The strength of this approach is its straightforward application, since the mean response depends on the covariates and not on any random effects or any previous responses. Thus, only the marginal distribution of the subject dependent vector is specified.

The variance of the response is a function of the mean and is conditional on the vector of covariates represented as

$$\mathrm{Var}(\mathrm{y}_{it} \mid \mathbf{x_{it}}) = v(\pi_{it})\phi$$

where $v$ is the variance function depending on $y_{it}$ and $\phi$ is the dispersion parameter assumed to be 1 for the exponential dispersion model family.

Let $D$ be a diagonal matrix of derivatives $\partial\pi_i/\partial\eta_i$ and $V(\pi_i)$ is a $n_i \times n_i$ diagonal matrix to be decomposed as;

$$V(\pi_i) = \mathbf{D}[V(\pi_{it})]^{\frac{1}{2}}\mathbf{I}_{(n_i \times n_i)}\mathbf{D}[V(\pi_{it})]^{\frac{1}{2}} \tag{5}$$

This estimation equation treats each observation within a given time point as independent. Our work focuses on the marginal distribution of the response for which the mean and the variance are averaged over the six observation time points. However, the variance of correlated data does not have a diagonal form and hence, we replace the identity matrix $\mathbf{I}_{(n_i \times n_i)}$ using methods proposed by Liang and Zeger [40] with another correlation structure $\mathbf{R}_i(\rho)$. $G_i$ is the diagonal matrix with $j^{th}$ the diagonal element equal to $v(\pi_{ij})$ such that Eq 5 corresponds to Eq 6

as shown:

$$W_i = \mathbf{G_i^{\frac{1}{2}}} \mathbf{R}_i(\boldsymbol{\rho}) \mathbf{G_i^{\frac{1}{2}}} \tag{6}$$

The working correlation structure $R_i$ with dimension $n_i \times n_i$ is assumed to depend on a vector of the association parameter $\rho$. Liang and Zeger [40] stated that the mis-specification of $\mathbf{R}_i(\boldsymbol{\rho})$ only affects the efficiency of the $\hat{\boldsymbol{\beta}}$ and $\hat{\boldsymbol{\beta}}$ is robust against mis-specification. In our application, we will consider several correlation structures. These include the unstructured structure, where every measure between two points is assigned its association parameter; the autoregressive (AR-1) structure with $lag = 1$, in which correlation decreases exponentially with the differences in measurements; the independence structure in which we use the identity matrix as the correlation structure; and the exchangeable structure in which correlation is assumed to be equal across different measurements. Liang and Zeger have provided evidence that misspecification of the correlation structure only affects $\beta$'s efficiency. This is because of the assumption that the estimation equation for the regression coefficients is orthogonal to the estimation equation for the correlation coefficients.

The GEE are as follows;

$$\sum_{1=1}^{j} \mathbf{D}_i^{\intercal} \mathbf{W}_i^{-1} (y_i - \pi_i) = \mathbf{0} \tag{7}$$

Where $D_i = \mathbf{G} \Lambda_i \, \mathbf{x}_i$, $W_i = V(\pi_{it})^{\frac{1}{2}} \mathbf{R}_i(\boldsymbol{\rho}) V(\pi_{it})^{\frac{1}{2}}$ and $\Lambda_i$ is a diagonal matrix with $j^{th}$ entry given by $\frac{dk^{-1}(\eta_{ij})}{d\eta_{lij}}$.

The most traditional way of solving the estimating equations is to employ the iterative reweighed least squares algorithm, which is a modification of the Newton–Raphson algorithm. In this approach, the observed Hessian matrix replaces the expected Hessian matrix, using the Fisher scoring algorithm.

However, McDaniel [41] proposed an alternative approach to estimate $\beta$'s such that instead of the summation in Eq 7, they are evaluated using the matrix form as shown;

$$\mathbf{x}^{\intercal} \Lambda \mathbf{G} \left( \mathbf{G^{\frac{1}{2}}} \mathbf{R}_i(\boldsymbol{\rho}) \mathbf{G^{\frac{1}{2}}} \right)^{-1} \mathbf{Z} \tag{8}$$

Several methods of analyzing skewed binary data have been proposed in the literature [9, 42–44]. Of particular importance for this current study is the method described by Prentice [5] that allows for the elimination of asymmetry through the modification of the inverse link function of the logit model, given as:

$$\left( \frac{\exp^{\mathbf{x}^{\intercal}\boldsymbol{\beta}}}{1 + \exp^{\mathbf{x}^{\intercal}\boldsymbol{\beta}}} \right)^{\alpha}$$

Statistical analysis was then implemented in *R* version 3.6.3 [45] (The R Development Core Team, Vienna, Austria). Though most functions are available directly in the software, we required an extra library including "dplyr" [46] for data manipulation, "glogis" [47] for the skewed logit Cumulative Density Functions (CDFs) plots with different values of $\alpha$ and the "geeM" [41] for the skewed logit analysis under the GEE.

The final GEE models were calculated and the probabilities of a child having morbidities were interpreted. These probabilities were calculated using the inverse-logit function and odds ratio as the exponential values of the differences in the logits. The *p*-values were calculated for each parameter estimate, as were the $Z$ statistic and the model and robust standard errors.

## Results

### Application to the real dataset and interpretation of findings

The preliminary analysis showed that 148 (45%) infants were born to women who tested positive for BV while the remaining 179 (55%) were born to women who tested negative for BV. 185 (57%) infants were breastfed, while 142 (43%) were formula-fed. 168 (51%) were males and 159 (49%) were female. 61 (19%) of the infants were HIV-positive, while 266 (81%) were HIV-negative.

It was of scientific interest to model the effects of BV on the marginal probability of an infant suffering from different morbidities in the first six months of life. Assuming morbidity incidence as the response, our data had the number of morbidity incidences recorded in a given month for each infant. Zero was recorded if no incidences occurred.

We sought to assess whether children born to women who tested positive for BV were more likely to have a higher morbidity incidence than their counterparts and if the effects would change with time. The literature has shown that BV has more effects during the first months after birth as the child continues to build immunity as they grow. Also, we expect children who gain weight consistently to have fewer morbidity incidences than those babies who take time to gain weight.

The frequency of morbidity incidence seemed to decrease evenly in the BV present group. This was not the same in the BV absent group, which evidenced increases and decreases in morbidities in the different months considered (Table 1). It was, therefore, important to examine the effect of BV on infant morbidity over time. In order to correctly estimate the marginal effect of the parameters of interest to be estimated, a distribution had to be chosen for the dependent variable, which did not involve assuming a specific distribution would apply. Thus, we considered the following logistic model:

$$
\text{logit}(\pi_{ik}) = \log\left(\frac{\pi_{ik}}{1 - \pi_{ik}}\right) = \beta_0 + \beta_1 \text{BV}_i + \beta_2 \text{HIV}_i + \beta_3 \text{feeding}_i + \beta_4 \text{male}_i
$$
$$
+ \beta_5 \text{time}_k + \beta_6 \text{weight}_i + \beta_{15} \text{time}_k \times \text{BV}_i
$$

(9)

for $k = 1, \ldots, 6$, $i = 1, \ldots, 327$ where $\text{Time}_k = k$. $\text{Male}_i = 1$ if the $i^{th}$ child is male and 0 if female, $\text{HIV}_i = 1$ if the child tests positive to HIV and 0 otherwise, $\text{Breastfed}_i = 1$ if the child was randomized to the breastfeeding group and 0 if randomized to the formula feeding group, $\text{BV}_i = 1$ if the mother tested positive for BV and 0 if she tested negative, and $\text{Weight}_i$ is recorded continuously for each infant across the six months. Data on morbidities from birth were included in the month 1 tally, and not as an independent time period, since morbidities due to BV on neonates was found to be insignificant in previous research [21, 32]. We interacted time and BV to assess changes in immunity over the time of exposure.

Our data can effectively account for the within-subject correlation. Hence, we consider the following correlations structures in terms of independence, as well as whether they are exchangeable, AR(1), M-dependent, and unstructured. In this paper, we were interested in comparing two models, the skewed logit-GEE and the standard GEE, when the response is assumed to be asymmetric. We assessed different correlation structures and all our parameter estimates were within the acceptable standard error ranges. However, measurements which were not taken for the same individual exhibit lower correlation and follow a pattern imposed by AR(1), thus for the interpretation of our work, AR(1) was adopted. For the M-dependent variable, we use the default $m = 1$.

Very few iterations are needed for the convergence of the models in GEE. Therefore, we initially set the maximum iterations to 50, however, the models with M-dependent variable and

AR(1) correlation structure did not converge. We increased the maximum iterations to 200 and this achieved convergence. More precisely, independence converged after 12 iterations, exchangeability was achieved after 16 iterations, unstructured compliance was achieved after 14 iterations, AR1 was achieved after 89 iterations and after 108 iterations, the model was *m*-dependent.

The results in Table 2 showed a significant differences in the coefficients and their marginal effect, particularly in the interaction terms. When we chose a *p*-value = 0.05 level of significance, parameter estimates from the standard GEE were not significant. In this case, only time and gender were significant. However, when using the skewed logit GEE, gender, time, BV, and the interaction between time and BV were significant.

## Model-based vs sandwich-based variance ratio

Table 3 shows the differences in variances from the model and the Huber sandwich estimate in which we sought to establish by what factor are they different. This was calculated using

$$V.R = \left(\frac{\text{Robust S.E}}{\text{Model S.E}}\right)^2$$

As expected, and confirmed by our results, the major differences between the model-based and empirical variance occur as a result of the independence correlation structure. The largest differences are in the estimated variance of the BV with the sandwich-based variance ratio. There are differences, but these do not have a notable influence on the variances. They are comparable within the correlation structure. The least variances differences are observed in the AR(1) correlation structure. This supports our choice for using the AR(1) correlation structure for model interpretation. This is because, for a correctly specified correlation, we expect the model and sandwich errors to be comparable, thus increasing the efficiency in the estimation of the $\boldsymbol{\beta}$'s.

## Effects of time on BV

We proceed and calculate the effects of BV across time on infants given by $exp(\hat{\beta}_1 + \hat{\beta}_{15}time)$ and reported in Table 4. This table shows that the effects of BV on morbidity tend to decrease with time from month 1 to month 5. For example, if we compare month 1 and month 5, we can conclude that at month 1, the OR of having morbidity incidences are 3.37 times higher for exposed than unexposed babies. At month 5, the OR decreases to 1.11 for exposed babies. At month 6, we observe a reverse causality, whereby the unexposed had higher OR for morbidities. This can be explained such that sick babies had more hospital visits and therefore, were treated for different illnesses, thus achieving a better health status in the long run. This leaves the BV unexposed group of babies vulnerable to other illnesses during growth, with minimal health intervention as they rarely sought medical attention. This is likely due to the non-threatening nature of the health conditions. With time, these could have led to an increase in illnesses experienced by infants in the unexposed group.

## Discussion

In the present study, we utilized the skewed logit technique under the GEE framework to analyze the risk factors associated with BV. We built on the existing contributions put forth by Nagler [4] and Liang and Zeger [40]. The model adopted in the present study is based on logistic regression, but modified assuming a parameter for skewness, to allow it to accommodate both symmetric and asymmetric responses.

**Table 2. Regression parameter estimates with model-based and empirical Standard Errors (SE) for independence, exchangeable, AR(1), unstructured and $M$-dependent correlation structures estimated using unconditional residuals for GEE and skewed logit-GEE.**

| Effect | Corr | GEE | | | | | SL-GEE | | | | |
|---|---|---|---|---|---|---|---|---|---|---|---|
| | | Est | Model SE | Rob SE | Wald Z | $p$-value | Est | Model SE | Rob SE | Wald Z | $p$-value |
| Intercept | Ind | 0.253 | 0.228 | 0.276 | 0.918 | 0.359 | 0.176 | 0.209 | 0.239 | 0.737 | 0.461 |
| | Exch | 0.088 | 0.263 | 0.264 | 0.335 | 0.738 | 0.024 | 0.242 | 0.235 | 0.100 | 0.920 |
| | AR(1) | 0.119 | 0.267 | 0.276 | 0.430 | 0.667 | 0.043 | 0.245 | 0.239 | 0.180 | 0.858 |
| | Unstr | 0.029 | 0.272 | 0.272 | 0.108 | 0.914 | -0.038 | 0.249 | 0.238 | -0.160 | 0.873 |
| | $M$-dep | 0.129 | 0.261 | 0.276 | 0.468 | 0.640 | 0.050 | 0.239 | 0.239 | 0.207 | 0.836 |
| Breastfed | Ind | -0.057 | 0.108 | 0.157 | -0.361 | 0.718 | -0.062 | 0.099 | 0.136 | -0.455 | 0.649 |
| | Exch | -0.022 | 0.146 | 0.150 | -0.148 | 0.883 | -0.027 | 0.134 | 0.134 | -0.205 | 0.838 |
| | AR(1) | -0.052 | 0.137 | 0.157 | -0.332 | 0.740 | -0.058 | 0.126 | 0.136 | -0.425 | 0.671 |
| | Unstr | -0.027 | 0.149 | 0.155 | -0.173 | 0.863 | -0.031 | 0.137 | 0.138 | -0.224 | 0.823 |
| | $M$-dep | -0.051 | 0.131 | 0.157 | -0.323 | 0.747 | -0.057 | 0.121 | 0.136 | -0.416 | 0.678 |
| BV | Ind | 1.086 | 0.431 | 0.791 | 1.373 | 0.170 | 1.495 | 0.348 | 0.420 | 3.561 | <0.001 |
| | Exch | 1.049 | 0.470 | 0.802 | 1.308 | 0.191 | 1.475 | 0.371 | 0.419 | 3.524 | <0.001 |
| | AR(1) | 1.000 | 0.533 | 0.910 | 1.099 | 0.272 | 1.494 | 0.421 | 0.444 | 3.368 | <0.001 |
| | Unstr | 0.901 | 0.530 | 0.957 | 0.941 | 0.347 | 1.286 | 0.440 | 0.553 | 2.326 | 0.020 |
| | $M$-dep | 1.017 | 0.526 | 0.926 | 1.098 | 0.272 | 1.513 | 0.417 | 0.451 | 3.353 | <0.001 |
| BV:Time | Ind | -0.199 | 0.091 | 0.145 | -1.376 | 0.169 | -0.275 | 0.077 | 0.083 | -3.310 | <0.001 |
| | Exch | -0.198 | 0.088 | 0.144 | -1.368 | 0.171 | -0.277 | 0.072 | 0.083 | -3.340 | <0.001 |
| | AR(1) | -0.191 | 0.107 | 0.163 | -1.176 | 0.240 | -0.280 | 0.088 | 0.087 | -3.214 | <0.001 |
| | Unstr | -0.176 | 0.099 | 0.170 | -1.036 | 0.300 | -0.246 | 0.084 | 0.103 | -2.383 | 0.017 |
| | $M$-dep | -0.196 | 0.106 | 0.166 | -1.180 | 0.238 | -0.285 | 0.088 | 0.088 | -3.227 | 0.001 |
| HIV | Ind | 0.189 | 0.179 | 0.309 | 0.612 | 0.541 | 0.222 | 0.159 | 0.244 | 0.910 | 0.363 |
| | Exch | 0.248 | 0.250 | 0.299 | 0.830 | 0.406 | 0.273 | 0.225 | 0.249 | 1.096 | 0.273 |
| | AR(1) | 0.216 | 0.234 | 0.326 | 0.662 | 0.508 | 0.253 | 0.208 | 0.253 | 1.001 | 0.317 |
| | Unstr | 0.193 | 0.253 | 0.323 | 0.596 | 0.551 | 0.232 | 0.225 | 0.259 | 0.894 | 0.371 |
| | $M$-dep | 0.212 | 0.224 | 0.328 | 0.646 | 0.518 | 0.250 | 0.198 | 0.253 | 0.989 | 0.323 |
| Male | Ind | -0.370 | 0.117 | 0.168 | -2.202 | 0.028 | -0.382 | 0.107 | 0.145 | -2.632 | 0.009 |
| | Exch | -0.358 | 0.156 | 0.158 | -2.261 | 0.024 | -0.358 | 0.144 | 0.143 | -2.495 | 0.013 |
| | AR(1) | -0.359 | 0.148 | 0.166 | -2.162 | 0.031 | -0.369 | 0.135 | 0.144 | -2.568 | 0.010 |
| | Unstr | -0.319 | 0.159 | 0.165 | -1.937 | 0.053 | -0.329 | 0.145 | 0.146 | -2.262 | 0.024 |
| | $M$-dep | -0.363 | 0.142 | 0.167 | -2.173 | 0.030 | -0.373 | 0.130 | 0.144 | -2.589 | 0.010 |
| Time | Ind | 0.178 | 0.062 | 0.076 | 2.346 | 0.019 | 0.192 | 0.057 | 0.066 | 2.915 | 0.004 |
| | Exch | 0.146 | 0.067 | 0.075 | 1.941 | 0.052 | 0.165 | 0.061 | 0.065 | 2.539 | 0.011 |
| | AR(1) | 0.135 | 0.071 | 0.076 | 1.783 | 0.075 | 0.150 | 0.065 | 0.065 | 2.289 | 0.022 |
| | Unstr | 0.110 | 0.069 | 0.075 | 1.457 | 0.145 | 0.126 | 0.063 | 0.065 | 1.918 | 0.055 |
| | $M$-dep | 0.143 | 0.070 | 0.076 | 1.875 | 0.061 | 0.156 | 0.064 | 0.066 | 2.370 | 0.018 |
| Weight | Ind | -0.112 | 0.057 | 0.071 | -1.581 | 0.114 | -0.125 | 0.052 | 0.061 | -2.043 | 0.041 |
| | Exch | -0.068 | 0.067 | 0.069 | -0.982 | 0.326 | -0.087 | 0.062 | 0.060 | -1.447 | 0.148 |
| | AR(1) | -0.062 | 0.067 | 0.071 | -0.871 | 0.384 | -0.076 | 0.062 | 0.061 | -1.242 | 0.214 |
| | Unstr | -0.034 | 0.068 | 0.072 | -0.481 | 0.631 | -0.050 | 0.063 | 0.062 | -0.805 | 0.421 |
| | $M$-dep | -0.068 | 0.065 | 0.071 | -0.953 | 0.341 | -0.080 | 0.060 | 0.061 | -1.311 | 0.190 |

There are several situations in which the relationship between the function of the response and covariates is not strictly symmetric. The asymmetric model is a class of models that borrows strength from both symmetric and asymmetric forms and can be applied in both scenarios, while still maintaining model parsimony. Furthermore, the frequently encountered

**Table 3. Differences in model-based vs sandwich-based variance ratios for both GEE and SGEE.**

| Effect | Corr | GEE | | SL-GEE | |
|---|---|---|---|---|---|
| | | Est | V.R | Est | V.R |
| Intercept | Ind | 0.253 | 1.465 | 0.176 | 1.317 |
| | Exch | 0.088 | 1.006 | 0.024 | 0.944 |
| | AR(1) | 0.119 | 1.062 | 0.043 | 0.952 |
| | Unstr | 0.029 | 0.996 | -0.038 | 0.917 |
| | M-dep | 0.129 | 1.115 | 0.050 | 0.997 |
| Breastfed | Ind | -0.057 | 2.099 | -0.062 | 1.893 |
| | Exch | -0.022 | 1.051 | -0.027 | 0.997 |
| | AR(1) | -0.052 | 1.309 | -0.058 | 1.169 |
| | Unstr | -0.027 | 1.085 | -0.031 | 1.022 |
| | M-dep | -0.051 | 1.433 | -0.057 | 1.274 |
| BV | Ind | 1.086 | 3.364 | 1.495 | 1.458 |
| | Exch | 1.049 | 2.915 | 1.475 | 1.275 |
| | AR(1) | 1.000 | 2.911 | 1.494 | 1.109 |
| | Unstr | 0.901 | 3.262 | 1.286 | 1.578 |
| | M-dep | 1.017 | 3.108 | 1.513 | 1.172 |
| BV:Time | Ind | -0.199 | 2.507 | -0.275 | 1.168 |
| | Exch | -0.198 | 2.690 | -0.277 | 1.331 |
| | AR(1) | -0.191 | 2.328 | -0.280 | 0.977 |
| | Unstr | -0.176 | 2.923 | -0.246 | 1.505 |
| | M-dep | -0.196 | 2.446 | -0.285 | 1.006 |
| HIV | Ind | 0.189 | 2.968 | 0.222 | 2.343 |
| | Exch | 0.248 | 1.436 | 0.273 | 1.225 |
| | AR(1) | 0.216 | 1.947 | 0.253 | 1.487 |
| | Unstr | 0.193 | 1.634 | 0.232 | 1.322 |
| | M-dep | 0.212 | 2.145 | 0.250 | 1.626 |
| Male | Ind | -0.370 | 2.057 | -0.382 | 1.845 |
| | Exch | -0.358 | 1.027 | -0.358 | 0.989 |
| | AR(1) | -0.359 | 1.268 | -0.369 | 1.128 |
| | Unstr | -0.319 | 1.069 | -0.329 | 1.001 |
| | M-dep | -0.363 | 1.395 | -0.373 | 1.234 |
| Time | Ind | 0.178 | 1.498 | 0.192 | 1.344 |
| | Exch | 0.146 | 1.261 | 0.165 | 1.137 |
| | AR(1) | 0.135 | 1.132 | 0.150 | 1.003 |
| | Unstr | 0.110 | 1.207 | 0.126 | 1.090 |
| | M-dep | 0.143 | 1.180 | 0.156 | 1.049 |
| Weight | Ind | -0.112 | 1.538 | -0.125 | 1.365 |
| | Exch | -0.068 | 1.057 | -0.087 | 0.963 |
| | AR(1) | -0.062 | 1.120 | -0.076 | 0.979 |
| | Unstr | -0.034 | 1.098 | -0.050 | 0.988 |
| | M-dep | -0.068 | 1.189 | -0.080 | 1.042 |

assumption of symmetry is very restrictive, unrealistic, and can lead to incorrect conclusions regarding the parameter estimates.

The model we have used in this study has been shown to be useful in applications when the symmetry properties of a binary outcome are unknown, and it seems to be applicable in both symmetric and asymmetric cases. Due to the correlated nature of longitudinal data, and

**Table 4. Calculated coefficient of bacterial vaginosis with time from $exp(\beta_1 + \beta_{51} \times time)$, achieved by replacing the respective values from the skewed logit-GEE model with the AR-1 correlation structure.**

| Time | Coefficient of effects of bacterial vaginosis |
|------|-----------------------------------------------|
| Month 1 | 3.37 |
| Month 2 | 2.54 |
| Month 3 | 1.92 |
| Month 4 | 1.45 |
| Month 5 | 1.11 |
| Month 6 | 0.83 |

needing an easy means of marginal interpretation, the GLM methods seem insufficient, but the use of GEE has been recommended and successfully applied in recent literature.

In this paper, we found that gender is a reliable predictor of infant morbidity. Specifically, girls were more likely to be healthy than boys. This finding is supported by previous studies and adds to the large body of knowledge indicating that boys require more attention and health care than girls. With girls having a higher survival probability than boys, our results appear consistent with the reports of Stevenson et al. [48]. This finding implies that there is hopes for a decline in mortality among boys if better interventions targeting their health can be implemented.

BV was found out to have a significant relationship with infant morbidities when other covariates are controlled for. Infants whose mothers tested positive for BV were found to have higher morbidity incidences compared to those whose mothers tested negative. The effect of BV on infant health has been reported in several studies, but with different conclusions on morbidities and mortalities [49, 50]. The most important finding in this work was the degree of significance observed in the skewed logit model for the interaction between BV and time. This finding would be of interest to doctors, as it indicates the need to plan for proper treatment and monitoring of an infant's health after confirming the maternal BV status, particularly during the first six months. This finding can also inform targeted infant morbidity campaigns, depending on the mother's BV status and the age of the infant.

The negative coefficient of weight and infant morbidities could indicate that an increase in weight gain could reduce morbidity. Babies who eat well tend to gain nutrients from food and have better capabilities of fighting illness in their bodies. Proper weight gain is also an indicator of proper growth. These results were consistent with those reported by Sarah *et al.* [51].

Past work by Verma *et al.* indicated an increase in the number of illnesses during infant growth [52]. This is in contrast with what was reported in Table 1, which shows only an insignificant decline among all the infants(5%), from a high of 61% to a low of 56%. To be more precise, considering the BV-exposed group, there was a huge decline of 25%, from a high of 78% in the first month to 53% in the sixth month. However, in the non-exposed group, there was a slow increase, whereby the number of morbidities observed increased from 47% in month one to 58% in month six. However, this study was based on the general population of the infants, without factoring in any other factors defining the exposed group or applying any randomization.

Not all covariates included in our study were statistically significant at $p = 0.05$. Nonetheless, their coefficient sign could assist in detecting a trend of association with the response. The covariate set included, e.g. the mode of feeding, whereby breastfeeding had a negative relation with infant morbidities. This finding could reflect behaviors that have been reported in other studies whereby breastfed infants were healthier than their counterparts who were formula-fed [32, 53]. Finally, the HIV status of the infant exhibited a positive coefficient with infant

morbidity. Infants who tested positive for HIV presented signs of morbidity, consistent with the results obtained by Kartik *et al*. [53]. Morbidities associated with HIV were found to increase infant mortality risk according to studies conducted in Kenya [54], Botswana [55], Cameroon [49], and South Africa [53].

To our knowledge, this is the first health research study that considers a skewed logit model under the GEE framework. Our study is one of the few studies that specifically explores the effect of BV on infants across time and considering the HIV status.

## Conclusion

Longitudinal binomial data are likely to be observed in numerous health fields where the binary components are correlated. Logit and probit models are widely used for modeling this outcome, which means applying the assumption that data is symmetrical. However, some competing methods for symmetry have been proposed as the logit and probit models do not support skewed binomial responses. We have shown that skewed logit-GEE is able to show an association between variables which is not identified by the standard GEE. Accordingly, it fits our imbalanced health dataset better. In this study, we have shown the superiority of the SL-GEE over the standard GEE when asymmetry is assumed. In our approach, the score of morbidities is converted to a binary, with asymmetry in the extreme morbidity cases. Literature supports an association between BV and morbidity among infants [21]. Thus, since skewed logit-GEE has predicted a BV-time interaction, we conclude that asymmetry is an important factor to consider before choosing the analysis method. It must be appropriately accounted for in analytical models to avoid biases in final parameter estimates, as has been established in this paper and other works [3–5, 11].

Our research has focused on the commonly neglected "minor diseases" which have been ignored at the expense of "major causes" of infant morbidity and mortality [56, 57]. Therefore, we recommend further research and policies that target infant morbidity on a more holistic level.

## Acknowledgments

We acknowledge the anonymous reviewers whose comments improved the quality of this paper significantly.

## Author Contributions

**Conceptualization:** Ngugi Mwenda.

**Data curation:** Ngugi Mwenda.

**Formal analysis:** Ngugi Mwenda.

**Methodology:** Ngugi Mwenda.

**Software:** Ngugi Mwenda, Gregory Kerich.

**Supervision:** Ruth Nduati, Mathew Kosgei, Gregory Kerich.

**Validation:** Gregory Kerich.

**Writing – review & editing:** Ngugi Mwenda, Ruth Nduati, Mathew Kosgei.

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
