## [Decision Letter · Decision Letter 0]

17 Aug 2020

PONE-D-20-14520

Time effects of bacterial vaginosis on infant morbidities in Kenya assessed using modified skewed generalized estimating equations

PLOS ONE

Dear Dr. Mwenda,

Thank you for submitting your manuscript to PLOS ONE. After careful consideration, we feel that it has merit but does not fully meet PLOS ONE’s publication criteria as it currently stands. Therefore, we invite you to submit a revised version of the manuscript that addresses the points raised during the review process.

We look forward to receiving your revised manuscript.

Kind regards,

Musie Ghebremichael

Academic Editor

PLOS ONE

Journal Requirements:

2. Please specify in your ethics statement whether participant consent was written or verbal. If verbal, please also specify: 1) whether the ethics committee approved the verbal consent procedure, 2) why written consent could not be obtained, and 3) how verbal consent was recorded.

4. Please ensure that you include a title page within your main document and list all authors and all affiliations as per our author instructions and clearly indicate the corresponding author.

Reviewers' comments:

Reviewer's Responses to Questions

**Comments to the Author**

1. Is the manuscript technically sound, and do the data support the conclusions?

Reviewer #1: No

Reviewer #2: Partly

Reviewer #3: Partly

2. Has the statistical analysis been performed appropriately and rigorously? 

Reviewer #1: Yes

Reviewer #2: Yes

Reviewer #3: No

3. Have the authors made all data underlying the findings in their manuscript fully available?

Reviewer #1: No

Reviewer #2: No

Reviewer #3: No

4. Is the manuscript presented in an intelligible fashion and written in standard English?

Reviewer #1: No

Reviewer #2: Yes

Reviewer #3: Yes

5. Review Comments to the Author

Reviewer #1: The main focus of this article is to implement a non-standard GLM/GEE method to analyze the infant morbidity and mortality, due to skewness of data. The authors incorporate Burr Type X (generalized Rayleigh) distribution and geeM package in R to achieve this goal.

My main concern is with the Methodology. I recommend that the authors focus on how the statistical analysis was implemented in geeM package of R, rather than going over the general technical details (see below). For this, the authors should understand the geeM package better, in order to make clearer and better technical arguments. In particular, the article will be more attractive if the authors include the R codes, either in the text or in the appendix/supplemental material.

For specific comments, there are instances where things could be eliminated or moved to the appendix. For example, the explanations from line 244 to 280 are standard in GLM and can be found in many books and articles, so the authors can just give a reference without having to explain in 3 pages.

Also, the notations are very inconsistent throughout. The both the X's and Y's are sometimes capital letters and sometimes lower case letters and sometimes bold letters; in both cases the subscripts are either inconsistent or non-existent. In addition, authors sometimes use X^T and other times X' to denote the X transpose. More seriously, the formulas (3) and (4), as well as the formula on line 309, are incorrect. These inconsistencies make the paper very difficult to read and make sense.

Again, the authors must take care to make the technical arguments coherent and sound. By focusing more on understanding and implementing the R package that actually perform the analysis, the hope is that the paper will become more consistent and practical.

Reviewer #2: See attached

Please use the space provided to explain your answers to the questions above. You may also include additional comments for the author, including concerns about dual publication, research ethics, or publication ethics. (Please upload your review as an attachment if it exceeds 20,000 characters) (Limit 200 to 20000 Characters)

Reviewer #3: This study uses data from the Nairobi infant morbidity study to evaluate the effects of time and bacterial vaginosis (BV) on morbidity among infants born to women infected with HIV-1. The authors compared the results of their data analysis using a traditional generalized estimating equations (GEE) model versus their newly-developed skewed generalized estimating equations (SGEE) model, which extends the GEE framework to accommodate possible asymmetry.

General comments

It is unclear if the authors are presenting a novel statistical method versus a novel application of an existing method to shed light on the effects of time and VB on infant mortality. Note that lines 182-184 suggest the former while lines 137-139 and 455-456 suggest the latter.

If the formulation of their SGEE model is novel, it would be helpful to formally validate the utility of this method (versus GEE) is the presence of asymmetry using simulation studies. If this work has already been done and published elsewhere, and the authors are presenting a novel application of this method, I’m wondering if it would be possible to exclude the majority of the technical information contained in the methods section (or at least move it to an appendix), and report a validated goodness of fit/model comparison measure for the GEE versus SGEE models applied to their data.

Specific comments

Introduction:

-Line 40: I’m not sure what is meant by “distorted” in this sentence. Do the authors mean “transformed”?

-Lines 42-44: “A typical assumption for the distribution of the error term in logistic analysis is the logistic function that is commonly applied to data with standard binomial distributions.” I believe the phrasing that suggests logistic regression models have error variables that follow a logistic distribution is characteristic of discrete choice models, primarily used by economists. Given that this paper involves a biomedical application (and will likely appeal to a clinical audience) it would be helpful to rephrase.

-Line 58: “The supporting literature has shown that this could be inefficient for parameter estimation in some settings” could use a supporting reference.

Methods:

-The notation introduced at the beginning of the methods section is ambiguous. For example, line 146 introduces “nj” as the number of observations observed for subject “i”. Later, the notation “ni” is used to denote the dimension of the covariate matrix and response vector. Repeated use of the same letter with varying subscripts is confusing. Elsewhere (line 150) the authors use Yij to (presumably) denote the jth observation for subject i, but then two lines later (line 152) define “Yi” as an indicator for subject i’s morbidity without reference to a particular time. This suggests that the authors are modeling a univariate response. Was the second subscript j left off in line 152?

-Line 149: I’m not sure what a matrix-vector is? Do they mean covariate matrix?

-Line 188: Y0 is not defined.

Results

-Line 342: It would be good to explicitly state how the response variable was categorized (e.g. 0 vs at least 1 morbidity at a given monthly visit)

-Line 352: Since the authors are interested in comparing morbidity by month among infants born to mothers with vs without BV, it would be easier to digest Table 1 if it presented the percentage of infants with at least one morbidity conditional on BV status during each month. For example, during month 1 a direct calculation of the 115/(115+33)=78% of infants experiencing a morbidity within the BV-present group vs the 85/(85+94)=47% of infants experiencing a morbidity within the BV-absent group highlights the relevant comparison more clearly.

-Line 357: “important to examine the effect of BV on child morbidity” should add “over time”

-Line 358: Was data from birth through 6 months used in this analysis (as stated in line 325)? Here t=1,2,…,6 suggests data from birth was not used.

-Line 364: “We wanted to show that our model fits better for binary data that violate symmetry”. As noted above, this general goal is better achieved using simulations.

-Line 365: “We tested different correlation structures for comparison purposes; however, because our model was concerned with the time effect, we used the AR(1) output for interpretation” This point could be clarified and could use some elaboration. How did the correlation structures compare overall? How is the use of AR(1) beneficial regarding modeling the effect of time in this application? Is it because measurements taken further apart have lower correlation according to the pattern imposed by AR(1)?

-Line 381: Where is -0.372 coming from? The coefficient for sex using the SGEE model with AR(1) is -0.369. Also, it would be helpful to explicitly note that this is an odds ratio (line 380).

-Line 381: “Those with BV had a 4.48 odds of morbidity compared with their counterparts” is this the OR at birth? I am not sure how this OR is interpreted because time is coded as t = 1, 2, ..,6 (358) and there is an interaction term. Did the authors use data at birth (time =0)?

-Line 386: The authors should explicitly state that the quantity they are using in an OR.

-Line 392 (and Table 3): Again, was data from birth used in this application or is the effect being extrapolated? (again, line 358 suggests outcome measurements were at months 1-6)

-Line 395: Where is the quantity 2.72 coming from? Isn’t the OR 3.37? And should 1.04 be 1.11?

Discussion

-Line 410: Best to avoid causal language such as “proved”.

-Line 419: Again, avoid using the word “proven”.

-Line 451: “In addition, our study found a decline in the number of morbidities from birth” Table 1 suggests that there was a decline in morbidity (defined as no morbidity vs at least 1 morbidity) over time among infants in the BV-positive group but an increase in morbidity over time among infants in the BV-negative group.

-Lines 461-463: “Asymmetry in the binary outcome is a phenomenon that should be appropriately accounted for in analytical models to avoid biases in final parameter estimates, as has been established in Table 2” It’s a stretch to draw this conclusion based on an inspection of p-values from just one application. The authors should cite simulation studies using their exact formulation of the SGEE model or validated measures of fit applied to this data.

-Line 464: I believe the term “converges” should be replaced with “is equivalent”

6. PLOS authors have the option to publish the peer review history of their article (what does this mean?). If published, this will include your full peer review and any attached files.

Reviewer #1: No

Reviewer #2: No

Reviewer #3: No

---

## [Author Response · Author response to Decision Letter 0]

28 Oct 2020

I have responded to all the reviewers comment on the letter attached in the file upload section

---

## [Decision Letter · Decision Letter 1]

16 Dec 2020

PONE-D-20-14520R1

Skewed-Logit model for Analyzing Correlated Infants morbidity data

PLOS ONE

Dear Dr. Mwenda,

Thank you for submitting your manuscript to PLOS ONE. After careful consideration, we feel that it has merit but does not fully meet PLOS ONE’s publication criteria as it currently stands. Therefore, we invite you to submit a revised version of the manuscript that addresses the points raised during the review process.

We look forward to receiving your revised manuscript.

Kind regards,

Musie Ghebremichael

Academic Editor

PLOS ONE

Reviewers' comments:

Reviewer's Responses to Questions

**Comments to the Author**

1. If the authors have adequately addressed your comments raised in a previous round of review and you feel that this manuscript is now acceptable for publication, you may indicate that here to bypass the “Comments to the Author” section, enter your conflict of interest statement in the “Confidential to Editor” section, and submit your "Accept" recommendation.

Reviewer #1: (No Response)

Reviewer #2: (No Response)

2. Is the manuscript technically sound, and do the data support the conclusions?

Reviewer #1: Yes

Reviewer #2: Partly

3. Has the statistical analysis been performed appropriately and rigorously? 

Reviewer #1: Yes

Reviewer #2: Yes

4. Have the authors made all data underlying the findings in their manuscript fully available?

Reviewer #1: No

Reviewer #2: No

5. Is the manuscript presented in an intelligible fashion and written in standard English?

Reviewer #1: Yes

Reviewer #2: Yes

6. Review Comments to the Author

Reviewer #1: For the GEE model, it would be helpful to show additional details of model selection. Authors may consider using QIC or QICu functions in R package MuMIn.

On page 18, equation (9), there should only be 6 main effects. I think beta_4 is redundant. Also, for the interaction term, it is more customary to write beta_{15} with (BV X time).

Again, providing the computer code will be very much appreciated.

Reviewer #2: My comments have been addressed except my second comment, beginning "The application of the scobit method to the data should include some justification based on the data, before seeing results ..." The authors reply in their letter with a justification for using scobit that is independent of the results of the data analysis. This argument, based on earlier medical analyses, satisfied me. However, I don't know whether they included this key justification in the revision of the paper; perhaps I missed it.

7. PLOS authors have the option to publish the peer review history of their article (what does this mean?). If published, this will include your full peer review and any attached files.

Reviewer #1: No

Reviewer #2: No

---

## [Author Response · Author response to Decision Letter 1]

18 Dec 2020

For the GEE model, it would be helpful to show additional details of model selection. Authors

may consider using QIC or QICu functions in R package MuMIn

Response: Thankyou for highlighting this. We have reviewed the methodology underlying QIC and QICu

framework as provided by Hardin (2013), in their book Generelized Estimating Equations. On page 163-171,

and they explain that the framework is good in model selection only when (1) a researcher is seeking to find the

best correlation structure among the competing ones and (2) when models have different number of covariates

and a researcher wants to choose the most parsimonous.

However, our approach is comparing the Logit and the Skewed Logit, where the later factors in binary

skeweness. In this scenario, both the QIC and the QICu are not appropriate for model selection. this is

because, in the book of Hardin (2013) on page 171, they point out that ”This criterion (QIC and QICu) is

meant as a guide for choosing between models when no scientific knowledge would guide the researcher to a

preference.” For our modeling approach, a large body of literature supports maternal Bacterial Vaginosis and

time effect on infant morbidity, therefore we prefer output from the skewed Logit since it was able to bring out

this important association.

On page 18, equation (9), there should only be 6 main effects. I think β4 is redundant. Also, for

the interaction term, it is more customary to write β15 with (BV ×) time

Response: Thank you for noting this. We have removed the redundant β4BVi to ensure we only have 6 betas.

Again, providing the computer code will be very much appreciated.

Response: We have anonymized the dataset and have provided it for replication purpose.As advised by

PLOS one, data have been shared and achieved in Fig share and can be accessed using the following link.

(https://osf.io/yfzw5) for data and r code at OSF public site

”The application of the scobit method to the data should include some justification based on the

data, before seeing results ...”

Response: Thank you for this. We have captured this in the Data section, the last paragraph that reads ” A

standard questionnaire developed by the principal investigator of the study to identify illnesses was completed

for both the mother and the child. This was achieved using a 19-item yes/no morbidity questionnaire which

purports to measure health status of an infant. The total score of the questionnaire is computed as the count of

all the ”yes” responses. There were a total of 1962 observations from 327 pairs of mothers and babies. From the

total score, we created a binary response of; (1) those who did not have any illness and (2) those who had either

minimal or severe illnesses. Table 1 presents an initial exploratory analysis used to identify the asymmetry in

the total responses for each month. This evidence of asymmetry justifies the use of the skewed logit

model.”

---

## [Editor Report · Decision Letter 2]

18 Jan 2021

Skewed-Logit model for Analyzing Correlated Infants morbidity data

PONE-D-20-14520R2

Dear Dr. Mwenda,

We’re pleased to inform you that your manuscript has been judged scientifically suitable for publication and will be formally accepted for publication once it meets all outstanding technical requirements.

Kind regards,

Musie Ghebremichael

Academic Editor

PLOS ONE

Additional Editor Comments (optional):

None
---

## [Editor Report · Acceptance letter]

21 Jan 2021

PONE-D-20-14520R2 

Skewed Logit Model for Analyzing Correlated Infant Morbidity Data  

Dear Dr. Mwenda:

I'm pleased to inform you that your manuscript has been deemed suitable for publication in PLOS ONE. Congratulations! Your manuscript is now with our production department. 

Kind regards, 

on behalf of

Dr. Musie Ghebremichael 

Academic Editor

PLOS ONE